# *Mycobacterium chimaera* Infections in a Unit of Cardio Surgery: Study at a General Hospital in Padua, Italy

**DOI:** 10.3390/microorganisms12010029

**Published:** 2023-12-23

**Authors:** Silvia Cocchio, Michele Nicoletti, Fabio Zanella, Dania Gaburro, Roberto Bianco, Gino Gerosa, Cristina Contessa, Margherita Boschetto, Paola Stano, Valentina Militello, Claudia Cozzolino, Tiziano Martello, Vincenzo Baldo

**Affiliations:** 1Department of Cardiac, Thoracic, Vascular Sciences, and Public Health, University of Padua, 35131 Padua, Italy; silvia.cocchio@unipd.it (S.C.); michele.nicoletti.md@gmail.com (M.N.); gino.gerosa@unipd.it (G.G.); claudia.cozzolino@studenti.unipd.it (C.C.); 2Cardiocirculatory Pathophysiology Unit, Surgery Department, University of Padua, 35131 Padua, Italy; fabio.zanella@aopd.veneto.it (F.Z.); dania.gaburro@aopd.veneto.it (D.G.); 3Department of Cardio Surgery, University of Padua, 35131 Padua, Italy; roberto.bianco@aopd.veneto.it; 4Department of Directional Hospital Management, University Hospital of Padua, 35128 Padova, Italy; cristina.contessa@aopd.veneto.it (C.C.); tiziano.martello@aopd.veneto.it (T.M.); 5Infection Control Division, University Hospital of Padua, 35128 Padova, Italy; margherita.boschetto@aopd.veneto.it; 6Microbiology and Virology Department, University Hospital of Padua, 35128 Padua, Italy; paola.stano@aopd.veneto.it (P.S.); valentina.militello@aopd.veneto.it (V.M.)

**Keywords:** *Mycobacterium chimaera*, cardio surgery, biosurveillance, HCUs, sustainable disinfection

## Abstract

*Mycobacterium chimaera* is a slow-growing non-tuberculous mycobacterium already known for being able to colonize cardio surgery heater–cooler units (HCUs). This study aims to describe the real magnitude of the phenomenon, providing a methodological protocol and the results of a longitudinal survey. In the period 1 January 2017–23 May 2022, over 1191 samples were collected on 35 HCUs of two different manufacturers. Among them, we identified 118 (10.3%) positive results for *M. chimaera*. We propose our 4-year biosurveillance experience as a practical model to minimize microbiological patients’ risk, suggesting the need for new procedures and interventions for a safer and more ecological cardio surgery.

## 1. Introduction

*Mycobacterium chimaera* is a slow-growing non-tuberculous mycobacterium (NTM) ubiquitously present in the environment, which can be an opportunistic human pathogen. It is intrinsically resistant to most classes of antibiotics and disinfectants thanks to its biofilm formation skill [1]. Indeed, NTMs are approximately 1000 times more resistant to chlorine than *Escherichia coli*, the bacterium used as a standard reference by the disinfection industry [2].

Its acknowledgment as a species within the *Mycobacterium Avium Complex* (MAC) happened in 2004, after a proposal by Tortoli [3]. In the beginning, *M. chimaera* had been linked to lung infections, especially in patients with underlying lung disease, such as chronic obstructive pulmonary disease. Only later was its name related to cardiac surgery: the first connection was reported in the spring of 2015 when the Swiss surveillance system reported a cluster of six patients with invasive infection with *M. chimaera*. All the infected individuals had previously undergone an open-heart surgery procedure. Investigators recognized the cause of infection in the common use of contaminated devices: the heater–cooler units (HCUs). HCUs are medical devices used for body temperature regulation, commonly employed during cardiopulmonary bypass surgery or in other procedures requiring extracorporeal circulation [4]. In these surgeries, the patient’s blood is cooled down and subsequently rewarmed by the heat exchanger integrated in the oxygenator. The proper functioning of the heater–cooler unit depends on the heat exchanger, which is generally water-based.

Before the *M. chimaera* era, this class of medical device never had a specific surveillance system for NTMs. However, it was already known that the presence of warmed water could create an optimal culture medium for microorganisms. To reduce the sedimentation of lime, rust and other inorganic substances, a frequent change of water has indeed always been indicated to reduce the risk of bacterial colonization [5].

After the first alert in 2015, a special Task Force was named by the European Centre for Disease Prevention and Control (ECDC). The group firstly investigated and retrieved any available data, but due to the lack of specific surveillance protocols, only a limited number of sources were available: the first epidemiological description of the phenomenon was provided, introducing the first operative actions [6,7]. The ECDC released a specific protocol for the identification of cases with laboratory tests, also recommending environmental tests, for which common methodological instructions were given [8]. The risk assessment identified the hazard in the aerosol produced by the colonized heater–cooler units: fans inside the unit could disseminate the NTMs within the airflow [9]. If contaminated aerosol could reach the surgical field, there could be a risk of contamination of the open wound as well as of the entire instrument table, especially for foreign material implants. Not all the patients who underwent open-chest surgery at infected Swiss facilities acquired the diseases, probably due to differences in settings and exposure times, but this is currently the most likely transmission mechanism among *M. chimaera* cases who received open-chest surgery [10].

At the end of 2016, the European ECDC presented a rapid risk assessment with new preventive measures, such as the relocation of HCUs outside the operating room [10]. Meanwhile, the producer companies tried to design new solutions to reduce patient risks. One of the most diffused heater–cooler unit models was structurally redesigned to definitely eliminate the problem of aerosolization by means of a vacuum and sealing system on the water tanks [11].

Finally, the Health Security Committee consisting of European Commission experts issued a statement to (i) facilitate the exchange of information between Member States; (ii) implement control measures throughout Europe; and (iii) coordinate the European authorities in the event of identification of new cases. From that moment, the Italian Health Ministry activated specific national surveillance for *M. chimaera* cases and of *M. chimaera*-contaminated devices [11,12,13].

Among the various heater–cooler unit brands, one of the devices mostly associated with *M. chimaera* infection was the Sorin Stöckert 3T (Sorin Group, Munich, Germany; currently LivaNova, Sorin Group Deutschland GmbH, Norderstedt, Germany) [14]. Multiple molecular epidemiological studies were conducted to describe the global outbreak of *M. chimaera*. Most of them agreed that contamination at the LivaNova plant could have been the greatest source of *M. chimaera* infections in patients who previously underwent open-heart surgery [1,15].

Thanks to van Ingen’s work, it is possible to compare the genome of new cases with the sequencing of *M. chimaera* previously isolated in Swiss, German, Dutch, British, American, and Australian cases.

In the last official report in 2019, the Italian Health Ministry referred to 36 confirmed cases, all linked to cardiac surgery. The first one was identified by the Padua Center in 2018. In Europe, the total was 120 [11,12]. These numbers probably critically underestimate the true size of the phenomenon: under the assumption of 300,000 global annual valve replacement surgeries in the 10 major markets, the annual incidence was estimated among 156–282 cases only in these 10 countries [16]. In the absence of international biosurveillance systems of *M. chimaera*, the actual lethality rate is hard to determine. Among the reported cases, it amounted to 58.3% [13]. Special precautions should be adopted to protect non-immunocompetent patients.

In conclusion, more than a decade after the first case, the nature of *M. chimaera* and its epidemiology are still largely unknown. The absence of diffused surveillance, as well as difficult diagnosis, nonspecific clinical presentation, and year-long clinical manifestations [17] are key elements on which the scientific international community is called upon to work. In this study, we present our local experiences in HCU mycobacterial contamination and the adopted systematic biosurveillance strategies.

## 2. Materials and Methods

### 2.1. Setting

Our center is a university tertiary-care public hospital featuring all medical specialties located in Padua, in the north-east of Italy. With its 1682 beds, it is currently one of the most important hospitals in Italy. The cardiac surgery unit is a regional and national reference for a range of surgical procedures: coronary surgery (with or without extracorporeal circulation aid, or minimally invasive approaches); valve repair surgery (traditional or minimally invasive techniques); valve replacement surgery (traditional, minimally invasive or transcatheter techniques); valve-sparing aortic root surgery (either replacement or conservative); conservative and radical surgery of heart failure; circulation mechanical assistance (either short-term or long-term, with traditional or minimally invasive approaches); treatment of congenital heart disease in adults (either by traditional therapy, minimally invasive or transcatheter methods); and heart transplant or mechanical assistance of circulation.

### 2.2. Surveillance System

The local Preventive Medicine, the Hospital Infection Management Committee and the Hospital Medical Direction cooperated to evaluate a periodical risk assessment.

We initiated a prospective surveillance program on all HCUs in use. Samples are scheduled monthly, and the microbiological exams are performed by the local Microbiology and Virology Unit. In case of contamination in the regular screening, we perform another disinfection of the device. After this, we perform a second test. For every contamination found after the disinfection procedure, the device is sent to the home factory for deep disinfection; otherwise, if the contamination is resolved by the disinfection procedure, we keep the device in use. All the patients exposed to devices for which there were pending exams have been listed. In case of deep disinfection failure, we immediately communicate this to the manufacturer, the regional authority, and the Italian Health Ministry.

The surveillance program began on 6 December 2016, and it is still ongoing. In this study, we restricted our analysis to the period from 1 January 2017 to 23 May 2022.

In our settings, heater–cooler units consist of tanks supplying water at controlled temperatures to heat exchangers and heating–cooling blankets through closed water circuits. The protocol consists of regular cleaning and disinfection of all the device components following the manufacturers’ indications and environmental testing. Patients exposed to contaminated devices were promptly reported to the local Preventive Departments.

### 2.3. Sampling Procedure and Microbiological Culture Procedure

A monthly surveillance of all hospital HCUs was implemented. Samplings were performed by the internal Epidemiological Service of the hospital, a unit made of specifically trained nurses. The samplings were originally performed as described by the manufacturer’s instructions on the two circuits that compose an HCU. The first one provides the cardioplegic solution to the heart while the second one is connected to the systemic circulation. Both were tested before and after the routine disinfection procedure. In case of failure (positivity after the regular disinfection), the device received a deep disinfection, after which it was re-tested. The two circuits are connected to single-use elements with a system called a Hansen quick-coupling joint which we discovered could also be contaminated. From that point, we improved the sampling strategy testing to include this joint.

Mycobacterial cultures were performed on 1 L of water samples, concentrated by a cellulose nitrate membrane (0.45 μm) and resuspended in 5 mL of 0.85% saline solution. Concentrated samples were decontaminated using BBL MycoPrep solution (Becton Dickinson, Franklin Lakes, NJ, USA) and resuspended in 2 mL of phosphate-buffered solution. Then, 500 μL of decontaminated samples were inoculated into bottles of Middlebrook 7H9 Liquid Medium, as required by the manufacturer, prior to OADC and PANTA antibiotic mixture inoculation. Meanwhile, a Middlebrook 7H10 was also sown with the same volume. Plates were incubated for 6–8 weeks at 35–37 °C until positivity.

A sample was declared negative when bacteria were undetectable over 6–8 weeks [18]. The analysis of the isolations was performed by one of the two microbiologist experts in mycobacteria. Indeed, all mycobacteria identified as *M. intracellulare* using the GenoType Mycobacterium CM line probe assay kit (HAIN Lifescience/Arnika, Nehren, Germany) were further confirmed as belonging to the *M. chimaera* species using the GenoType NTM-DR kit (HAIN Lifescience/Arnika).

We also collected the HCUs’ airflow with a Surface Air System Super ISO USB 100 produced by PBI International (Houston, TX, USA) (currently VWR), a microbiological air sampler, at a rate of 100 L/min. Middlebrook 7H11 agar plates were used. About 0.5 m^3^ of air was sampled both from the front (at 30 cm) and the back of each device. Tests were performed on devices that were running for at least 5 min. Mycobacterial cultures performed on airflow samples were inoculated on selective Middlebrook 7H11 plates and incubated at 35 °C.

To discover the source of recontamination in some devices, we introduced an internal strategy: in addition to the recommended tests, we also performed sampling on the Hansen quick-coupling joint which connects the HCUs to the oxygenators. This sample site was not initially included in regional protocols.

When samples could not be immediately analyzed, the Microbiology Unit stored them for at least 24 h, either between 2 °C and 8 °C for water, or at room temperature for airflow.

Each sample was labeled with the date of collection, instrument serial number, site collection and telephone number for any communications.

### 2.4. Disinfection Procedures

In Stockert 3T devices, LivaNova recommends surfaces and water circuits are disinfected before first use, before device storage, and during its regular use. We performed surface disinfections after every use; water replacement (adding 100 mL 3% hydrogen peroxide to tanks) and overflow bottle disinfection were performed every 7 days; water circuit disinfection every 14 days; and tube replacement every year. Complete cleaning and disinfection were also annually carried out by the manufacturer. A dedicated line was installed to provide softened and filtered water. Water filters were changed monthly and regular samplings were performed on the tap. The disinfection procedure was performed with Puristeril^®^ 340 (Fresenius Medical Care AG & Co. KGaA, Bad Homburg, Germany) at a final concentration of 3.3% every 14 days. In case of a failure of the disinfection procedure, we considered the HCU as affected by a persistent contamination, which meant that the LivaNova devices were shipped to the main factory for a deep disinfection [19].

Cleaning and disinfection of the Maquet HCUs instead includes a daily check of water level and water circulation in the condenser; weekly internal circulation cleaning (using 2% chloramine-T or 5% chloramine-T solution when atypical mycobacteria were found in the water system); and annual (or after 1000 h of use) device revision by the manufacturer [20]. In addition to the recommended disinfection procedures, since September 2019 we have performed a monthly disinfection treatment of the Hansen coupling connectors that consists of one hour of submersion in a 2% chloramine-T solution.

## 3. Results

In the period under examination, 1 January 2017 to 23 May 2022, we performed 1191 samplings; 1145 (96.1%) were carried out on HCU water tanks and 46 (3.8%) on HCU airflows. Overall, 35 HCUs were analyzed: 16 were produced by LivaNova and 19 by Maquet. The distribution of samples among brands was as follows: 384 samplings (33.5% of 1145) were performed on LivaNova HCUs (HCUs type 1), and 761 samplings (66.5% of 1145) were performed on Maquet HCUs (HCUs type 2). On both the brands, sampling was performed on the water tanks (“Myc”) and on the Hansen quick-coupling joint (“Hansen”), respectively, 298 (77.6%) and 86 (22.4%) times in the LivaNova HCUs and 546 “Myc” (71.7%) and 215 “Hansen”(28.3%) in the Maquet ones.

The estimated overall service period of HCUs under the monitoring program was 28,878 days. From the beginning of the study, sampling was performed on average every 25 days. Specifically, the number of days of use for type 1 and 2 HCUs were 6945 and 21,933, respectively, with an average sample every 18 and 29 days (Table 1).

Overall, 118 positive tests for *M. chimaera* (10.3%) were identified: 83 on type 1 HCUs (21.6%) and 35 on type 2 HCUs (4.6%). Furthermore, 11 water samples (0.9%) were positive for *Mycobacterium gordonae*, 61 (5.1%) were positive for *Pseudomomas Aeruginosa* and 1 was positive for *Enterococci.* Unspecifiable bacterial growths, with colony counts ranging between 1 and 300, were also found in 201 other water samples. For type 1 and type 2 HCUs, respectively, Myc analysis found positive results in 41.4% and 2.9% of the tests performed on pre-disinfection samples compared to 15.9% and 8.6% in post-disinfection samples. Instead, with Hansen analysis, respectively, 14.6% and 1.9% of pre-disinfection samples and 0.0% and 0.0% of post-disinfection samples were positive. The number of tests and positivity rates for each stage and HCU brand are shown in Figure 1. In the chart, the “Maquet joint”, reported under the “other” category, represents the first sample carried out in the suspicion that *M. chimaera* could also be found in the quick joints of colonized HCUs (on 6 December 2016). From that point, in agreement with the Hospital Infection Management Committee, an additional monthly sanitization procedure of the Hansen quick joints was initiated (even if not foreseen by the suppliers). It should be noted that since the introduction of this further procedure, all the joint swabs have shown negative results post-disinfection.

Table 1 shows the positivity rate by HCU brand and ID number, regardless of the time of surveillance. Most of the criticalities were highlighted on type 1 instruments (21.6% versus 4.6% of the type 2 HCUs). Figure 2 shows *M. chimaera* screening outcomes pre- and post-disinfection, as required by manufacturer instructions for use (IFU) and by the regional legislation (Regional Council Resolution no. 999 of 12 July 2019). Results were grouped by HCU device type and by month of collection. Please note that in the period between April and May 2020, no samplings were recorded due to the suspension caused by the SARS-CoV-2 pandemic. In type 1 HCUs, except for July 2021, there was a positivity persistence in all the months in which at least one sample was collected. We found a much smaller positivity rate in type 2 HCUs (21.6% vs. 4.6%). The same result was confirmed in pre- and post-disinfection samplings (34.4% vs. 2.6% in pre-disinfection and 12.8 vs. 6.3% in post-disinfection). Detailed numbers are reported in Appendix A.

## 4. Discussion

The data suggest that both type 1 and type 2 heater–cooler units (HCUs) were easily colonized by *M. chimaera*. Among the two devices and their specific disinfection procedures, there appear to be some distinct technological features that result in higher positivity rates in type 1 HCUs. Although we have not been able to identify the critical element or elements, our surveillance confirms the existence of the problem; it might refer to a failure of a disinfection procedure or to an intrinsic HCU component. A previous study suggests the hypothesis of false negativizations after the initial contamination. Schreiber et al. suggest that negative results might indicate a too-low concentration of *M. chimaera* in the water tanks during the supposed biofilm creation [21]. Our experience partially sustains this theory, because nowadays we have not been able to identify an alternative source of recontamination of the deep-disinfected HCUs. Meanwhile, we remember that we identified a possible root of the recontamination processes in the Hansen quick-coupling joint. Indeed, we do not know if our strategy recognized all the possible sources of recontamination, but it was capable of successfully preventing any clinical cases over more than 5 years of surgical procedures using 35 devices.

The insight gained from this surveillance underscores that every medical process carries inherent risks for both patients and healthcare operators, as surfaces or devices can be susceptible to colonization. Certain technical features may contribute to elevated rates of colonization, prompting the need for specific risk management strategies. An illustrative example involves relocating devices outside surgical rooms unless strictly essential to the procedure.

If the current state of preventive medicine is characterized by the ability to generate international rapid risk assessments with a vast amount of data, a significant future objective should encompass the capacity to reduce risks even in unpredictable events and hazards. In our study, the dissemination of low-growing pathogens such as *M. chimaera* serves as a compelling case study. We explored the introduction of heater–cooler units, their use without sufficient concern and the subsequent awareness of an emerging hazard, along with the initial counteractions. Presently, we envision new technological opportunities and solutions, recognizing the indispensable role of biosurveillance systems in monitoring the evolution of rare events. Awareness and capability stand as essential elements of our Preventive Medicine Unit. Perhaps for this reason, we successfully identified the first confirmed case of *M. chimaera* in Italy.

We propose a dynamic surveillance system in which the sampling frequency can be adjusted based on the availability of new evidence regarding emerging hazards. Consequently, we conducted more analyses in the initial months when information on *M. chimaera* was limited. The same strategy was adopted in air sampling; as we documented that the sealed water tanks were not more capable of spreading *M. chimaera*, we provided an update in the risk analysis and in the sampling schedule. Indeed, our present emphasis is on the post-disinfection phase.

The attainment of absolute risk absence proves to be an unfeasible objective. Instead, it is imperative to assess individual patient risks. For instance, microorganisms may be present in drinking water [5], yet the general population remains healthy. This underscores the need for more detailed risk reduction programs, particularly when dealing with vulnerable patients such as heart transplant recipients or those receiving prosthetics. Generally, individuals with compromised immune systems or those receiving contaminated foreign tissues are at a heightened risk of developing comorbidities.

Therefore, meticulous recording of surgical procedures is essential to trace potential unknown hazard exposures. Emerging technologies, including new software or AI, hold promise in assisting preventive medicine by identifying small, unrevealed clusters. This underscores the significance of biosurveillance systems in achieving high-level healthcare. Although such technology is not yet available, we hope for its development as soon as possible. Its implementation would prove immensely beneficial for preventive medicine, the Hospital Infection Management Committee, and the Hospital Medical Direction.

Another aspect we aim to address pertains to the economic perspective. Currently, we are unable to provide a precise account of our direct and indirect costs, and cost analysis was beyond the scope of this work. The literature on the costs of nosocomial infections already exists [22], although not focused on *M. chimaera*, but we can imagine it as a normal healthcare-associated infection (HAI). So, our emphasis is more on the imperative to develop novel technologies incorporating antimicrobial features, not solely for cost reduction but also to curtail the use of detergents with high environmental impact. Moreover, both HCU types 1 and 2 necessitate complex disinfection procedures, typically conducted by expert technicians in specialized settings, such as the manufacturer’s headquarters.

Recently, a new waterless technology for HCUs has become available, although our direct experience with it is limited. While the water-free solution may enhance the antimicrobial properties of HCUs, a comprehensive assessment is imperative before any judgment. We intend to experiment with this solution to gain confidence and conduct a thorough risk assessment, focusing on patient safety, cost reduction and usage mitigation of environmental toxic compounds during routine and deep disinfection procedures.

## 5. Conclusions

In hospital settings, especially during delicate cardio-surgery procedures, the attainment of zero risk is an unfeasible goal. Our experience underscores the failure of the disinfection procedures prescribed by the manufacturers’ IFU but also highlight the need for a biosurveillance system coupled with the capability for rapid risk assessment, particularly in the context of rare and unforeseen events. This is crucial because isolated cases may be indicative of broader clusters that have hitherto gone unnoticed. Upon reaching this level of surveillance, a substantial volume of data can be generated, providing experts, e.g., engineers, with the foundation to develop updates and new technological features.

Currently, our heater–cooler unit (HCU) issues have been addressed with practical solutions: they are now positioned outside the surgical room, and the water tanks are securely sealed. In addition, each surgical intervention is meticulously linked to the devices used, facilitating the tracking of potential patient exposure to new hazards.

More studies on time- and cost-effective alternatives to standard disinfection, disassembly, replacement of parts and reassembly of HCUs are ongoing [23]. Further research is essential to explore low-dosage or chlorine-free chemicals and to find modern solutions with reduced environmental impact.

## Figures and Tables

**Figure 1 microorganisms-12-00029-f001:**
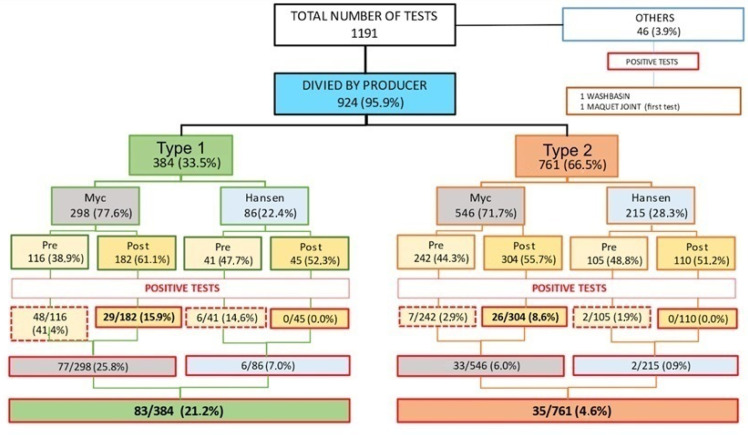
Flow-chart of the type of analysis, and positivity rate by device manufacturer.

**Figure 2 microorganisms-12-00029-f002:**
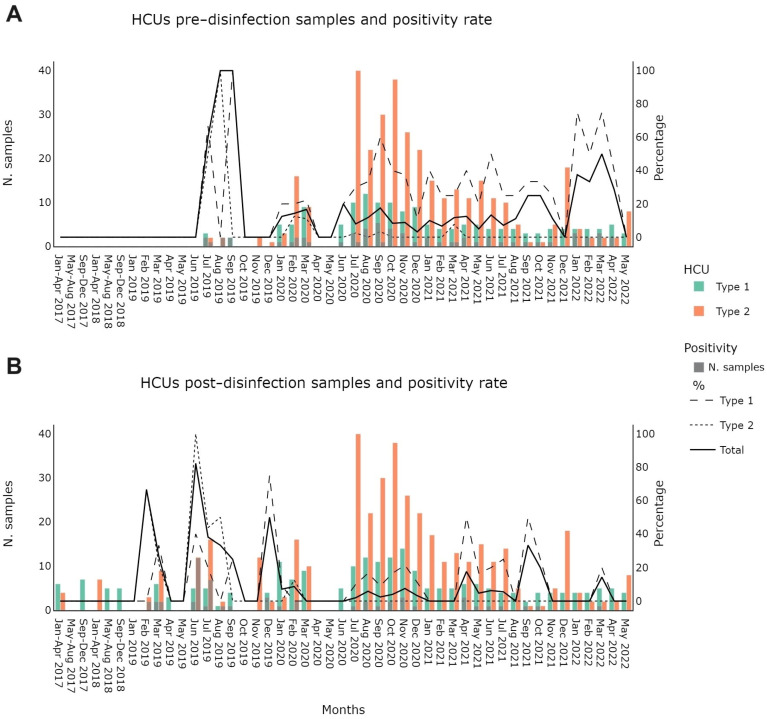
Pre- (**A**) and post- (**B**) disinfection number of samplings and positivity rate by month and HCU type.

**Table 1 microorganisms-12-00029-t001:** Distribution of service days from the start of the program, number of tests and positivity rates by HCU device.

HCU Type 1						HCU Type 2					
Machine ID	Days in Use	N. of Tests	Days/Tests Performed	Positivities		Machine ID	Days in Use	N. of Tests	Days/Tests Performed	Positivities	
				N.	%					N.	%
1	252	2	126	0	0.0	1	503	57	9	1	1.8
2	252	2	126	0	0.0	2	153	10	15	0	0.0
3	55	7	8	1	14.3	3	1994	31	64	3	9.7
4	961	27	36	3	11.1	4	797	50	16	1	2.0
5	841	39	22	12	30.8	5	749	46	16	0	0.0
6	868	38	23	11	28.9	6	749	42	18	0	0.0
7	826	9	92	4	44.4	7	749	42	18	0	0.0
8	106	3	35	1	33.3	8	731	42	17	1	2.4
9	286	42	7	6	14.3	9	585	26	23	0	0.0
10	525	45	12	10	22.2	10	585	20	29	0	0.0
11	420	51	8	16	31.4	11	172	4	43	0	0.0
12	223	29	8	7	24.1	12	1994	32	62	3	9.4
13	536	39	14	5	12.8	13	1994	32	62	6	18.8
14	341	21	16	0	0.0	14	1994	36	55	3	8.3
15	243	15	16	5	33.3	15	1994	63	32	4	6.3
16	210	15	14	2	13.3	16	1994	59	34	2	3.4
						17	1994	61	33	6	9.8
						18	1408	58	24	5	8.6
						19	794	50	16	0	0.0
Total	6945	384	18	83	21.6	Total	21,933	761	29	35	4.6

## Data Availability

The data supporting the findings of this study are available from the corresponding author upon reasonable request, and first must be approved by the Department of Directional Hospital Management of the University Hospital of Padua.

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
