# Peer review of "Mycobacterium chimaera Infections in a Unit of Cardio Surgery: Study at a General Hospital in Padua, Italy"

_microorganisms, 2023, doi:10.3390/microorganisms12010029_

Round 1

Reviewer 1 Report

Comments and Suggestions for Authors

Cocchio et. al. reported the results of a surveillance programme for the colonisation of cardio surgery heater-cooler units (HCUs) by Mycobacterium chimaera. Samples were collected between 2017 and 2022 and a comparison was between two types of HCUs was reported. Despite persistent colonisation of the units observed during the study period, no cases of M. chimaera infection were reported as a result.   

Major comments:

2.3 Sampling procedure

The authors should clarify who “the Epidemiological Service” and “the Microbiology Unit” are – this is not clear to an outside reader.

It is not clear what “patient and cardioplegia circuits” are – please clarify.

More detail about the sampling procedure is required, including how the samples were concentrated and the volumes that were added to the 7H9 liquid and 7H10 solid media.

Line 150: It should be indicated how bacteria were detected.

The manufacturer and model of the microbiological air sampler should be indicated and more detail about how this samples were analysed should be included.

Details about how the Hansen joints were sampled and analysed should be included.

2.4 Disinfection procedures

Line 178: Please define “persistent contamination”. This statement also contradicts the statement in lines 132 – 133.

2.5 Microbiological procedures

I would suggest that this section be combined with 2.3 for better flow.

Line 195: Details about how mycobacteria were initially identified should be included.

Results:

The authors should clarify the discrepancy between the number of water tank and air-flow samples, i.e., when were air-flow samples included and why was this not done for all the units?

Lines 203 – 206: “Myc” and “Hansen” analysis should be defined prior to use here. The authors should indicate the numbers of “Myc” analyses that also had “Hansen” analyses. The bullets can be removed so that these lines are included in the preceding paragraph.

Table 1: This table is not required as the information presented here is largely repeated in Table 2 and Figure 2.

Lines 214 – 226: Were any other mycobacteria detected? If so, this should be indicated. In addition, no mention of the heterotrophic bacteria is made here (but mentioned in methods). Was the level of contamination for either mycobacteria or heterotrophic bacteria recorded?

Figure 1: The “N.” should be removed from the boxes in the figure. The word “timing” should be removed from the title, as no timing is indicated in this figure.

Tables S1 and S1: replace “statistics” with “numbers”, as no statistical analysis was performed. It is not clear what the red values in these tables indicate.

Figure 2: Legend should appear below the figure. Please change “samplings” to “samples”. It is currently difficult to relate the sample numbers to the positivity rates in this figure. I would recommend that this be changed to the same scale, i.e. that it be represented as total number of samples and number of positive samples (rather than positivity %) for easier comparison. This could be represented as a two-colour bar graph.

The two types of units were disinfected in different ways. Did the authors investigate if the difference in positivity rates was due to the different disinfection procedures used?

Discussion:

The discussion is wordy and does not not discuss the data presented. This section should be reworked, and the data should be discussed in the context of previous epidemiological studies, such as  https://doi.org/10.1016/j.jhin.2022.11.003 ; DOI: 10.1055/s-0042-1756630

Minor comments:

Line 99: should read “10 major markets”.

Lines 101 and 251: italics for species name

Line 132: should read “contamination was found”.

Line 133: replace “;” with comma.

Line 134: replace “gave communication” with “communicated this”.

Line 136: replace “system” with “programme”.

Line 142: delete “or outbreak”.

Line 162: replace “conserved” with “stored”.

Line 175: add “out” before “by”.

Line 199: should read “under examination”.

Comments on the Quality of English Language

The manuscript would benefit from language editing.

Reviewer 2 Report

Comments and Suggestions for Authors

After verification as a separate species in 2004, M. chimaera came to the fore in 2015 when this pathogen was identified as the culprit in two long-term outbreaks of prosthetic valve and associated systemic infection in the United States and Europe. Since 2017, according to PubMed, 30 reviews have been published on the issue of M. chimaera, but the issue has not lost its relevance to date. In terms of pathogenicity, M. chimaera is transmitted during artificial blood circulation through bioaerosols released from contaminated water systems of heaters and coolers. Due to non-specific symptoms and a long latent period, post-operative infections caused by M. chimaera cannot be diagnosed and cured in time and can become life-threatening. In their publication, the authors confirmed the population of M. chimaera bacteria in two types of heater-cooler. The authors found that certain technical features can contribute to an increase in the colonisation rate of M.chimaega, which necessitates the development of specific risk management strategies and the use of more effective aseptic methods in heater-cooler operations. Although the reviewed article has limited novelty and theoretical value, it can be useful in solving a specific practical task, namely the prevention of infections caused by M.chimaera.
